# Uralenol, Glycyrol, and Abyssinone II as potent inhibitors of fibroblast growth factor receptor 2 from anti-cancer plants: A deep learning and molecular dynamics approach

Alomgir Hossain[ID][1,2]*, Md. Sanowar Hossan[1], Md. Shahanur Prodhan[1], Md. Nahid Hasan Joy[ID][1], Muntasir Rahman Siam[1], Md. Ekhtiar Rahman[1,2], Mohammad Nurul Matin[ID][2,3]*

1 Computational Biosciences and Chemistry Research Organization, Rajshahi, Bangladesh,
2 Department of Genetic Engineering and Biotechnology, University of Rajshahi, Rajshahi, Bangladesh,
3 Department of Biotechnology, Yeungnam University, Gyeongsan, Republic of Korea

* nmatin@ru.ac.bd (MNM); alamgir199817@gmail.com (AH)

## Abstract

Fibroblast Growth Factor Receptor 2 (FGFR2) plays a critical role in cellular proliferation and differentiation, and its dysregulation is associated with multiple cancers. This study integrates molecular docking, deep learning, pharmacokinetic profiling, and molecular dynamics (MD) simulations to identify potential FGFR2 inhibitors from a library of 1,350 phytochemicals derived from 51 anti-cancer medicinal plants that were traditionally used for anticancer purposes. Initial screening through AutoDock Vina revealed several top candidates with high binding affinities to FGFR2. The top three compounds, uralenol, glycyrol, and abyssinone II, underwent further evaluation via deep learning models, which predicted the potential efficacy of the pIC$_{50}$ (negative logarithm of the half-maximal inhibitory concentration) values. The ADME/T (absorption, distribution, metabolism, excretion, and toxicity) analysis confirmed favorable pharmacokinetic profiles and low toxicity risks. MD simulations validated the stability and compactness of protein–ligand complexes, with principal component analysis (PCA) and free energy landscape analyses confirming these interactions' conformational stability and thermodynamic favorability. These findings suggest that uralenol, glycyrol, and abyssinone II are potential FGFR2 inhibitors and need further experimental validation for potential therapeutic use in cancer treatment.

## Introduction

Globally, cancer is still a major and expanding public health concern. The latest global estimates (from GLOBOCAN 2022) indicate approximately twenty million new cancer cases and 9.7 million deaths in 2022 [1]. Even with major improvements in treatment methods, including surgery, radiation, and chemotherapy, cancer is still one

**Data availability statement:** All relevant data are within the manuscript and its Supporting Information files.

**Funding:** The author(s) received no specific funding for this work.

**Competing interests:** The authors have declared that no competing interests exist.

of the world's top causes of death [2]. Recent developments in cancer therapy have highlighted the efficacy of combinatorial approaches, including targeted medicines or conventional chemotherapeutics like taxanes and platinum compounds. Combining treatments increases effectiveness by targeting important pathways in concert, lowering drug resistance, and enhancing therapeutic outcomes, including tumor growth suppression and apoptotic induction [3]. The developmental processes from early embryogenesis to organogenesis depend on fibroblast growth factors and their corresponding receptors, fibroblast growth factor receptors, which are crucial in cellular signaling [4]. Four different isoforms of the FGF (fibroblast growth factor) and FGFR families-FGFR1, FGFR2, FGFR3, and FGFR4-are implicated in a variety of biological processes in mammals. FGFRs are receptor tyrosine kinases that are characterized by a single transmembrane domain that, when occupied by a ligand, promotes the activation of subsequent intracellular signaling cascades [5].

FGFR2 is an essential member of the receptor tyrosine kinase family that controls tissue development, differentiation, and proliferation, among other cellular functions [6]. The development and upkeep of various tissues and organs throughout the body depend heavily on these signaling pathways [7]. On chromosome 10q26, the FGFR2 gene has 20 exons in humans and 19 in mice. FGFR2b and FGFR2c are the two isoforms that it encodes via alternative splicing. FGFR2 is a membrane-bound receptor that has three domains: an intracellular tyrosine kinase domain, a transmembrane domain, and an extracellular ligand-binding domain [8–12]. FGF ligands are specifically bound to the extracellular domain of FGFR2, which initiates intracellular signaling cascades. The activation of FGFR2 results in the phosphorylation of several downstream signaling proteins, including transcription factors and adaptor molecules. Following this phosphorylation event, four key intracellular signaling pathways are activated: the PLCγ pathway, the JAK-STAT pathway, the PI3K-AKT-mTOR system, and the RAS-RAF-MEK-MAPK pathway [13,14]. These signaling networks are essential for controlling key cellular functions such as cell specialization, growth inhibition, programmed cell death, and cell division [15]. FGFR2 regulates cell proliferation, differentiation, and survival; its dysregulation is linked to developmental disorders and diverse cancers, making it a critical therapeutic target in oncology and regenerative medicine [16,17].

Dysregulation of FGFR2 signaling has been linked to several illnesses, such as endothelial-to-mesenchymal transition, cancer, chronic kidney disease, chronic obstructive pulmonary disease, respiratory distress syndrome, skeletal abnormalities, and craniosynostosis [18]. FGFR2 is a prominent target for therapeutic interventions due to its critical involvement in both normal physiology and disease pathogenesis. Understanding its function, structure, and regulation is essential for elucidating its role in development, disease, and potential therapeutic strategies [19,20]. FGFR2 is often overexpressed in a variety of malignancies. Well-known FGFR2 inhibitors, including erdafitinib, zoligratinib, AZD4547, infigratinib, BGJ398, dovitinib, and lucitanib, work by blocking FGFR2 activity, which suppresses tumor growth [8,21,22]. Regardless of their importance, these inhibitors' effectiveness is constrained by issues like a lack of specificity, off-target binding,

and drug resistance [23]. Three-dimensional quantitative structure-activity relationship, molecular docking, and dynamics studies are performed to address the challenges with medications in this class and to obtain knowledge about the structural requirements to develop more effective drugs.

Traditional drug discovery is costly, time-consuming, and linked to high attrition rates, which restricts patients' access to reasonably priced therapies [24]. Pre-clinical drug discovery relies heavily on bioinformatics, which uses a variety of tools and computational methods to maximize results [25]. This strategy focuses on improving the efficacy and selectivity of currently available medications and drug-like substances to find new therapeutic applications while minimizing the development time and expenses by employing proven safety profiles [26]. An integrated bioinformatics strategy is presented in this work to identify potential therapeutic candidates that target FGFR2.

## Materials and methods

### Protein preparation

The three-dimensional (3D) structure of FGFR2 was retrieved from the Protein Data Bank [27] (PDB ID: 6LVK) with a resolution of 2.29 Å. Before molecular docking studies, the protein structure underwent a series of preparatory steps. Initially, hydrogen atoms were added to account for any missing atoms from the structure. Subsequently, co-crystallized ligands and water molecules were removed to prevent interference during docking simulations. Finally, energy minimization and optimization were performed using the GROMOS 43B1 force field implemented in Swiss-PdbViewer v4.1.0 [28] to ensure the protein was in a suitable conformation for subsequent docking analyses.

### Ligand preparation

In this study, a comprehensive literature review was carried out to collect a list of 51 anti-cancer properties of medicinal plants that were traditionally used for anticancer purposes, with their 1350 phytochemicals (S1 Table) identified based on GC/MS (gas chromatography/mass spectrometry) analysis and downloaded as a structure data file (SDF) format from the PubChem database. To prepare these structures for molecular docking studies, OpenBabel v2.4.1 [29] was employed to convert the SDF files into a modified version of SDF, the PDBQT [Protein Data Bank, Q (charge), and T (AutoDock atom type)] format. Subsequently, the ligands underwent energy minimization using the MMFF94 force field and steepest descent optimization method with 2,000 iterations in PyRx software [30] to ensure the compounds achieved their most stable conformational states prior to molecular docking studies.

### Molecular docking

In drug discovery, molecular docking is essential for gaining a thorough understanding of the intricate molecular interactions underlying essential biological functions [31]. In this study, we performed molecular docking analyses on a compound library to discover potential inhibitors targeting FGFR2. The Autodock Vina screening tool [32] was employed to execute structure-based virtual screening by using preprocessed structures of all the phytochemicals and the receptor in the PDBQT file format. A structurally blind search was used in the virtual screening process, allowing the compounds to freely move and investigate their binding site or sites on FGFR2. For docking the compounds with PDB ID: 6VLK, the grid box was centered at coordinates X: 26.294, Y: 12.604, Z: 58.945, with dimensions set to X: 51.353, Y: 66.934, and Z: 59.605 Å. The ligand binding affinity was expressed in kcal/mol units as negative binding energy values [30]. The 40 highest-scoring ligands were chosen based on their optimal binding affinities. The re-docking procedure was conducted identically for FGFR2 proteins, and known inhibitors previously bound to the protein structures underwent re-docking to benchmark their binding affinities against our screened phytochemicals. Ultimately, 16 compounds were selected based on their most favorable negative binding affinities with the target protein for additional screening using deep learning methodologies.

### Re-screening using deep learning (DL)

The compounds with a binding affinity of greater than 9.0 kcal/mol were chosen for re-screening using a deep learning approach that used Recurrent Neural Networks (RNN) to predict models using the CHEMBL4142 dataset [33], which contained FGFR2 experimental inhibitors. This process was carried out using the DeepScreening server, which displays good screening performance with average and median area under the ROC (receiver operating characteristic) curve values of 0.86 and 0.89 under several built-in deep-learning models [34]. Using the PubChem fingerprint, PaDEL [35] produced 881 molecular fingerprints. Hyperparameters such as the hidden layer (3), number of neurons (2000, 700, 200), and learning rate (0.0001) were manually set for this developed model. The hidden layers employed the activation function, while the output layer utilized the sigmoid function. The predicted $pIC_{50}$ values were computed using the following formula, which was utilized to re-evaluate the 16 highest-scoring docked compounds and display the $pIC_{50}$ values:

$$pIC_{50} = -\log_{10}IC_{50}(M) = 9 - \log_{10}IC_{50}(nM).$$

The DeepScreening platform provides an integrated framework for deep learning–based virtual screening. Its workflow involves: (i) dataset preparation, where a specific target is selected to train the deep neural network (DNN); (ii) feature selection, in which molecular descriptors are identified and vectorized; (iii) model parameterization, where essential parameters are defined for training regression models; and (iv) virtual screening, in which the trained model is applied to evaluate large chemical libraries. DeepScreening is a fully automated and advanced server that combines data preprocessing, model construction, and virtual screening into a seamless pipeline. The deep learning model is limited by dataset size, chemical space representation, and potential bias in molecular descriptors, which may restrict the generalizability of predicted bioactivity.

### ADME/T analysis

After molecular docking, the ADME/T profile of the selected compounds was further evaluated to predict their drug-likeness properties. ADME/T stands for absorption, distribution, metabolism, excretion, and toxicity, which represents a crucial aspect of pharmaceutical development as it determines how a drug compound behaves within the biological system, like how a compound is absorbed into the bloodstream, distributed across tissues, metabolized by enzymes, eliminated from the body, and whether it has potential toxic effects. The SwissADME was used for pharmacokinetic and physicochemical property predictions, while ProTox-III was employed for toxicity assessment [36], with SMILES (simplified molecular-input line-entry system), a text-based representation of chemical structures, serving as the input format for the molecular structures.

### Molecular dynamics simulation

MD simulations were conducted on the three highest-scoring compounds, along with the control compound and Apo, to assess the stability of their corresponding protein-ligand complexes. The best-performing ligands underwent comprehensive all-atom MD simulations utilizing GROMACS version 5.1.5 for thorough analysis [37]. The simulation system was configured using LiGRO [38], a graphical user interface (GUI)-based application that streamlines the preparation workflow for MD simulations in GROMACS. Topological parameters for both the protein and ligand molecules were created employing the Amber99SB force field through the AnteChamber Python Parser interface (ACPYPE) [39], respectively. Ligand charge assignments were derived using the General Amber Force Field in conjunction with the bond charge correction approach. The protein–ligand complexes were solvated in a cubic simulation box using the TIP3P water model to mimic the aqueous environment. Subsequently, the systems were neutralized by introducing an appropriate number of counterions, and the ionic strength was adjusted by adding NaCl to a final concentration of 0.15 M. To eliminate potential steric clashes within the system, a two-stage energy minimization was conducted. Initially, 5000 steps were executed using the

steepest descent algorithm, followed by further refinement with the conjugate gradient method until the system's energy converged below 10 kJ/mol. Periodic boundary conditions were applied uniformly across all simulation boxes in all three spatial dimensions to replicate an infinite system and eliminate edge effects. The simulations were performed using NVT [number of particles (N), volume (V), and temperature (T)] and NPT [number of particles (N), pressure (P), and temperature (T)] ensembles, each running for 1 nanosecond duration, with temperature maintained at 310.15 K and pressure held at 1 bar. During both equilibration phases, positional restraints were applied to the heavy atoms of the protein and ligand with a force constant of 1000 kJ/mol/nm$^2$. Production molecular dynamics simulations were performed under NPT conditions with temperature maintained at 310.15 K using the V-rescale thermostat and pressure kept at 1 bar using the Parrinello-Rahman barostat. Non-bonded electrostatic and van der Waals forces were calculated using a cutoff radius of 1 nm. Bond constraints were applied to all heavy atoms utilizing the LINCS algorithm [40]. Subsequently, MD simulations were carried out for a duration of 100 nanoseconds (ns), employing a timestep of 2 femtoseconds, with trajectory frames recorded every 500 steps. The frames were stored after every 10 picoseconds (ps), and the trajectories were visualized using the PyMOL molecular graphics system, version 2.5, Schrödinger, LLC.

## Principal component analysis (PCA)

To investigate the overall conformational variability among the protein-ligand complexes, PCA was applied, incorporating structural comparisons with both the apo (unbound) protein and a standard drug-bound complex [41]. This multivariate technique enabled the identification and categorization of conformational shifts by evaluating variations across multiple structural descriptors throughout the MD simulations. PCA was performed through the diagonalization of covariance matrices, followed by the computation of eigenvalues and eigenvectors. The eigenvalues quantified the magnitude of atomic fluctuations, while the eigenvectors delineated the dominant directions of motion within the complex structures. Before PCA, the 100-nanosecond MD trajectories underwent pre-processing, including mean subtraction and normalization to unit variance, to standardize the data [42]. The analysis was implemented using Python (version 3.11), with the Scikit-learn library (version 1.2) facilitating the PCA computation, and Matplotlib (version 3.7) employed for data visualization.

## Free energy landscape

The free energy landscape (FEL) delineates the potential energy profile of a molecular system as a function of pertinent collective variables [43], providing essential insights into the energetic barriers and thermodynamically favorable conformational states that the molecule can occupy during transitions between distinct structural configurations [44]. The MD simulation trajectories underwent analysis through computation of root-mean-square deviation (RMSD) and radius of gyration (Rg) parameters for individual frames. These data were subsequently used to construct a density matrix, representing discrete regions within the conformational space. The Gibbs free energy surface was then derived from this density matrix through the application of statistical mechanics principles [45]. Two-dimensional and three-dimensional visualizations of the free energy landscapes were generated using Matplotlib version 3.7.2.

## Ethical consideration

This study does not require formal ethical approval, as it will rely solely on published data.

## Results

### Molecular docking analysis

Molecular docking is a computational technique used to predict the interaction between small molecules and macromolecular targets [46]. This method helps identify binding interactions, key binding residues, and binding affinity scores. In this study, docking protocols began with the selection of medicinal plants known for their anticancer

properties, identified through an extensive literature review. A phytochemical compound library containing 1,350 molecules was then constructed by collecting GC-MS data of these plants from previously published studies. The target protein in this study belongs to the transferase family and has a well-defined catalytic active site that was targeted to inhibit its function [47]. The binding affinities of the phytochemicals were calculated using AutoDock Vina and ranked based on their binding energy, where less negative scores indicate stronger binding [48]. A lower (i.e., more negative) binding score suggests a higher likelihood of stable ligand–protein complex formation, as it implies lower energy is required for bond formation [49]. To validate the docking protocol, re-docking was performed to assess consistency and accuracy.

Following the analysis of protein–ligand interactions, 16 phytochemical compounds were identified with favorable binding affinities. These include atherospermidine, β-hydroxyacteoside, uralenol, glycyrol, 2',4'-Dihydroxy-2''-(1-hydroxy-1-methylethyl)dihydrofuro[2,3-h]flavanone, broussoflavonol F, isolicoflavonol, vitexin, (2S)-euchrenone A7, abyssinone II, 4-[(3S,3aR,6S,6aR)-6-[3-(3,4-dihydroxyphenyl)-2-(hydroxymethyl)-2,3-dihydro-1,4-benzodioxin-6-yl]-1,3,3a,4,6,6a-hexahydrofuro[3,4-c]furan-3-yl]benzene-1,2-diol, prasterone, corosolic acid, lupeol, α-Amyrin acetate, and Fgfr3-IN-2. Their binding affinities (in kcal/mol) were: −9.6, −9.7, −9.8, −9.7, −9.3, −9.7, −9.4, −10.1, −10.4, −9.7, −10.0, −9.9, −10.1, −10.7, −10.6, and −9.0, respectively (Table 1). The docking scores predicted $IC_{50}$ values, and types of non-covalent interactions with the protein are summarized in Table 1. The reliability of the re-docking procedure was further confirmed through conformational superimposition of the ligand binding modes within the target protein (Fig 1). The calculated RMSD between the re-docked pose and the co-crystallized native ligand (inhibitor Fgfr3-IN-2) was 0.023Å, which is well enough below the commonly accepted 2Å threshold, thereby demonstrating the robustness of the docking protocol. This minimal deviation indicates that the binding pose was accurately reproduced, with the inhibitor occupying nearly identical sites and engaging the same key residues as in the native complex.

Based on binding affinity, interactions with active site residues, ADME profiles, and toxicity predictions, three compounds, uralenol, glycyrol, and abyssinone II, emerged as the most promising inhibitors (Fig 2). Uralenol showed a binding affinity of −9.8 kcal/mol and formed three hydrogen bonds with the residues Ala567, Glu534, and Ala567. Additionally, it formed six Pi-Alkyl interactions with residues Phe492, Leu487, Val495, Ala515, Leu633, and Ala643. Glycyrol exhibited a binding affinity of −9.7 kcal/mol. It formed one hydrogen bond with Ala567, one carbon–hydrogen bond with Gly588, and one Pi–Pi stacked interaction with Phe492. Moreover, it showed six alkyl interactions involving Ala515, Met538, Ile548, Val564, Lys517, and Val562. Abyssinone II demonstrated a binding affinity of −9.7 kcal/mol, forming three hydrogen bonds with residues Lys517, Ala567, and Glu534. In addition, it engaged in five Pi-Alkyl interactions with key hydrophobic residues, including Phe492, Leu487, Val495, Val564, and Leu633. These interactions suggest strong binding stability at the FGFR2 active site, reinforcing its potential as a promising FGFR2 inhibitor. The two-dimensional and three-dimensional interaction diagrams of uralenol, glycyrol, and abyssinone II with their control compound are demonstrated in Fig 3.

## Deep screening via deep learning

The deep learning approach employed artificial neural network frameworks to analyze $IC_{50}$ datasets and predict molecular bioactivity. A regression model was built using $IC_{50}$ values from 891 known FGFR2 inhibitors, utilizing PubChem fingerprints and standard neural network settings. The model demonstrated robust performance, achieving an $R^2$ value of 0.68, indicating a good fit between predicted and experimental $pIC_{50}$ values. Other evaluation metrics—MSE, RMSE, and MAE—were also low, confirming high predictive accuracy (Fig 4). The model was used to evaluate the bioactivity of selected compounds against the FGFR2 proteins. Uralenol, glycyrol, and abyssinone II showed $pIC_{50}$ values of 5.21, 5.06, and 4.66, respectively, suggesting potential inhibitory activity.

**Table 1. Molecular docking results of selected phytochemicals with FGFR2, including binding affinities, predicted pIC$_{50}$ values, interacting residues, interaction types, and bond distances.**

| Compound name and CID | Predicted pIC$_{50}$ score | Binding affinity (kcal/mol) | Residue in contact | Interaction type | Bond distance in Å |
|---|---|---|---|---|---|
| Atherospermidine (77514) | 5.32 | −9.6 | ALA567 | Hydrogen bond | 2.6674 |
| | | | ASP644 VAL495 | Carbon-hydrogen bond Pi-sigma | 2.83552 2.70035 |
| | | | PHE492 | Pi-alkyl | 4.91475 |
| | | | PHE492 | Pi-alkyl | 3.94792 |
| | | | ALA515 | Pi-alkyl | 4.71288 |
| | | | LEU633 | Pi-alkyl | 4.99796 |
| | | | LEU487 | Pi-alkyl | 4.52485 |
| beta-Hydroxyacteoside (10009317) | 5.05 | −9.7 | ASN631 | Hydrogen bond | 2.61246 |
| | | | GLU534 | Hydrogen bond | 2.38406 |
| | | | GLU574 | Hydrogen bond | 3.03798 |
| | | | ASP644 | Hydrogen bond | 2.2479 |
| | | | GLY570 | Carbon-hydrogen bond | 2.60764 |
| | | | ARG630 | Carbon-hydrogen bond | 2.56284 |
| | | | LEU487 | Carbon-hydrogen bond | 2.81792 |
| | | | ASP644 | Carbon-hydrogen bond | 2.66161 |
| | | | LYS517 | Pi-cation | 4.0545 |
| | | | MET538 | Pi-sulfur | 5.86616 |
| | | | LEU487 | Alkyl | 3.84378 |
| | | | VAL495 | Pi-alkyl | 5.26831 |
| | | | LYS517 | Pi-alkyl | 4.24801 |
| | | | VAL564 | Pi-alkyl | 4.14719 |
| 5315126 Uralenol | 5.21 | −9.8 | ALA567 | Hydrogen bond | 2.23283 |
| | | | GLU534 | Hydrogen bond | 2.71702 |
| | | | ALA567 | Hydrogen bond | 1.80355 |
| | | | PHE492 | Pi-alkyl | 4.43404 |
| | | | LEU487 | Pi-alkyl | 4.48572 |
| | | | VAL495 | Pi-alkyl | 5.33038 |
| | | | ALA515 | Pi-alkyl | 4.9365 |
| | | | LEU633 | Pi-alkyl | 4.49291 |
| | | | ALA643 | Pi-alkyl | 5.12151 |
| Glycyrol (5320083) | 5.06 | −9.7 | ALA567 | Hydrogen bond | 2.22708 |
| | | | GLY488 | Carbon-hydrogen bond | 2.62568 |
| | | | PHE492 | Pi-Pi T-shaped | 5.55891 |
| | | | ALA515 | Alkyl | 4.18778 |
| | | | MET538 | Alkyl | 5.15944 |
| | | | ILE548 | Alkyl | 5.0542 |
| | | | VAL564 | Alkyl | 3.87359 |
| | | | LYS517 | Alkyl | 3.53646 |
| | | | VAL562 | Alkyl | 5.03667 |
| | | | VAL564 | Alkyl | 3.55742 |
| | | | VAL495 | Pi-alkyl | 4.67316 |
| | | | LEU633 | Pi-alkyl | 4.67391 |
| | | | LEU487 | Pi-alkyl | 4.60825 |

*(Continued)*

| Compound name and CID | Predicted pIC$_{50}$ score | Binding affinity (kcal/mol) | Residue in contact | Interaction type | Bond distance in Å |
|---|---|---|---|---|---|
| 2',4'-Dihydroxy-2''-(1-hydroxy-1-methylethyl)dihydrofuro[2,3-h] flavanone (10291777) | 4.98 | −9.3 | ALA567 | Hydrogen bond | 2.40298 |
| | | | TYR566 | Carbon-hydrogen bond | 2.58771 |
| | | | LEU487 | Pi-sigma | 2.8667 |
| | | | VAL495 | Alkyl | 4.27032 |
| | | | PHE492 | Pi-alkyl | 4.7025 |
| | | | ALA515 | Pi-alkyl | 4.85932 |
| | | | LEU633 | Pi-alkyl | 4.57024 |
| Broussoflavonol F (9866908) | 4.9 | −9.7 | ASP644 | Hydrogen bond | 2.49049 |
| | | | ASP527 | Hydrogen bond | 2.09424 |
| | | | ARG664 | Pi-cation | 3.9714 |
| | | | ASP530 | Pi-anion | 4.13017 |
| | | | PHE492 | Pi-Pi stacked | 5.45939 |
| | | | LYS517 | Alkyl | 4.72981 |
| | | | LEU519 | Alkyl | 5.4929 |
| | | | LEU531 | Alkyl | 3.72256 |
| | | | LEU647 | Alkyl | 4.83729 |
| Isolicoflavonol (5318585) | 4.89 | −9.4 | ALA567 | Hydrogen bond | 2.18853 |
| | | | GLU534 | Hydrogen bond | 2.85995 |
| | | | PHE492 | Pi-sigma | 2.49293 |
| | | | VAL495 | Alkyl | 5.29702 |
| | | | LEU487 | Pi-alkyl | 4.38735 |
| | | | ALA515 | Pi-alkyl | 4.91865 |
| | | | LEU633 | Pi-alkyl | 4.54284 |
| | | | VAL564 | Pi-alkyl | 5.45039 |
| | | | ALA643 | Pi-alkyl | 5.22827 |
| Vitexin (5280441) | 4.88 | −10.1 | ALA567 | Hydrogen bond | 2.30149 |
| | | | ASN631 | Hydrogen bond | 2.5061 |
| | | | ARG630 | Hydrogen bond | 1.94279 |
| | | | LEU487 | Pi-alkyl | 4.38365 |
| | | | LEU633 | Pi-alkyl | 4.96536 |
| | | | VAL495 | Pi-alkyl | 4.89905 |
| | | | ALA515 | Pi-alkyl | 4.33192 |
| | | | LYS517 | Pi-alkyl | 5.15293 |
| | | | ILE548 | Pi-alkyl | 5.4911 |
| | | | VAL564 | Pi-alkyl | 3.8859 |
| | | | ALA643 | Pi-alkyl | 4.96609 |
| (2S)-Euchrenone A7 (44593508) | 4.66 | −10.4 | ASP644 | Hydrogen bond | 2.6215 |
| | | | LEU487 | Hydrogen bond | 2.16646 |
| | | | LYS517 | Pi-cation | 4.39463 |
| | | | VAL495 | Alkyl | 4.80041 |
| | | | PHE492 | Pi-alkyl | 3.8755 |
| | | | VAL564 | Pi-alkyl | 4.0897 |

*(Continued)*

**PLOS One**

**Table 1.** (Continued)

| Compound name and CID | Predicted pIC$_{50}$ score | Binding affinity (kcal/mol) | Residue in contact | Interaction type | Bond distance in Å |
|---|---|---|---|---|---|
| **abyssinone II (10064832)** | 4.66 | −9.7 | LYS517 | Hydrogen bond | 1.97204 |
| | | | ALA567 | Hydrogen bond | 1.96375 |
| | | | GLU534 | Hydrogen bond | 2.60461 |
| | | | PHE492 | Pi-alkyl | 4.38546 |
| | | | LEU487 | Pi-alkyl | 3.64809 |
| | | | VAL495 | Pi-alkyl | 4.64043 |
| | | | VAL564 | Pi-alkyl | 5.45831 |
| | | | LEU633 | Pi-alkyl | 5.2923 |
| **4-[(3S,3aR,6S,6aR)-6-[3-(3,4-dihydroxyphenyl)-2-(hydroxymethyl)-2,3-dihydro-1,4-benzodioxin-6-yl]-1,3,3a,4,6,6a-hexahydrofuro[3,4-c]furan-3-yl]benzene-1,2-diol (44243159)** | 4.32 | −10 | ALA567 | Hydrogen bond | 2.02747 |
| | | | ARG630 | Hydrogen bond | 2.21173 |
| | | | ALA567 | Hydrogen bond | 2.36634 |
| | | | ASP522 | Hydrogen bond | 2.28043 |
| | | | GLY570 | Carbon-hydrogen bond | 2.56627 |
| | | | PHE492 | Pi-Pi stacked | 5.61101 |
| | | | LEU487 | Pi-alkyl | 4.82118 |
| | | | VAL495 | Pi-alkyl | 5.43404 |
| | | | ALA515 | Pi-alkyl | 4.90992 |
| | | | LEU633 | Pi-alkyl | 4.30529 |
| **Prasterone (5881)** | 3.31 | −9.9 | ASN571 | Hydrogen bond | 2.67403 |
| | | | VAL495 | Alkyl | 4.11563 |
| | | | ALA515 | Alkyl | 5.21458 |
| | | | LYS517 | Alkyl | 5.02817 |
| | | | VAL564 | Alkyl | 5.27687 |
| | | | LEU633 | Alkyl | 4.95413 |
| | | | ALA643 | Alkyl | 4.66463 |
| | | | PHE492 | Pi-Alkyl | 4.64879 |
| **Corosolic acid (6918774)** | 3.2 | −10.1 | GLY570 | Carbon-hydrogen bond | 2.50904 |
| | | | LEU487 | Alkyl | 5.25131 |
| | | | VAL495 | Alkyl | 4.70336 |
| | | | ALA515 | Alkyl | 5.1973 |
| | | | LEU633 | Alkyl | 4.45376 |
| | | | VAL564 | Alkyl | 4.80933 |
| | | | PHE492 | Pi-alkyl | 5.16668 |
| **Lupeol (259846)** | 2.68 | −10.7 | VAL495 | Alkyl | 5.01783 |
| | | | ALA515 | Alkyl | 3.7026 |
| | | | VAL564 | Alkyl | 5.0194 |
| | | | ALA567 | Alkyl | 3.90632 |
| | | | LEU633 | Alkyl | 4.67136 |
| | | | ALA643 | Alkyl | 4.87092 |
| | | | LYS517 | Alkyl | 4.53334 |
| | | | VAL564 | Alkyl | 4.70208 |
| | | | ARG630 | Alkyl | 3.85471 |
| | | | LEU487 | Alkyl | 3.7966 |
| | | | PHE492 | Pi- alkyl | 5.34853 |

*(Continued)*

**Table 1.** (Continued)

| Compound name and CID | Predicted pIC$_{50}$ score | Binding affinity (kcal/mol) | Residue in contact | Interaction type | Bond distance in Å |
|---|---|---|---|---|---|
| **alpha-Amyrin acetate (92842)** | 2.58 | −10.6 | LYS520 | Hydrogen bond | 2.73412 |
| | | | LEU487 | Alkyl | 4.89901 |
| | | | VAL495 | Alkyl | 4.55629 |
| | | | ALA515 | Alkyl | 4.68412 |
| | | | ARG630 | Alkyl | 5.22533 |
| | | | LEU633 | Alkyl | 4.53066 |
| | | | VAL564 | Alkyl | 4.09089 |
| | | | PHE492 | Pi-alkyl | 5.20643 |
| **Fgfr3-IN-2 (146018253)** | 5.14 | −9 | ASP522 | Carbon-hydrogen bond | 2.6655 |
| | | | LEU487 | Carbon-hydrogen bond | 2.6193 |
| | | | ASP644 | Carbon-hydrogen bond | 2.69076 |
| | | | PHE492 | Pi-Pi stacked | 4.37262 |
| | | | ALA643 | Alkyl | 5.15135 |
| | | | LEU647 | Alkyl | 2.2223 |
| | | | ARG664 | Alkyl | 4.6649 |

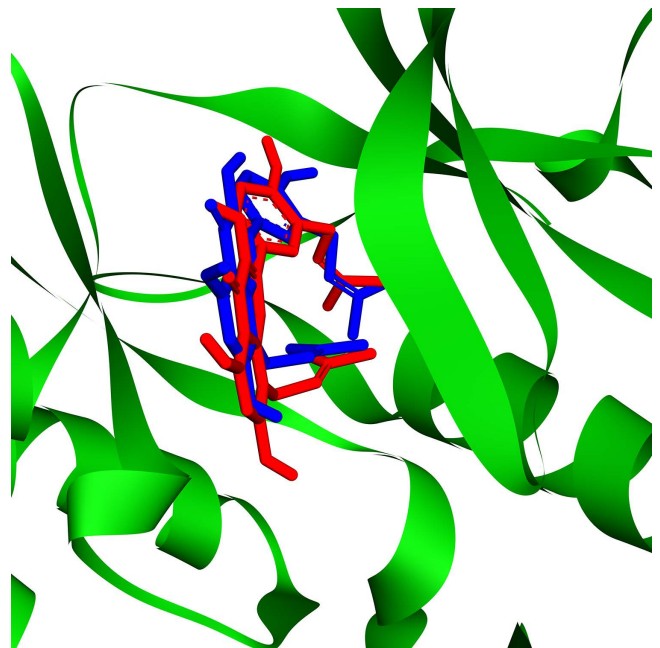

**Fig 1. Conformational superimposition of the ligand.** Three-dimensional visualization of the best re-docking pose of FGFR2, showing the conformational superimposition of the co-crystallized ligand (inhibitor Fgfr3-IN-2). The calculated RMSD of 0.023Å indicates excellent agreement between the re-docked and native binding conformations.

## Pharmacokinetics and toxicity analysis

Pharmacokinetics refers to the study of the dynamic processes that govern the movement of chemical substances throughout the body. It encompasses the kinetic behavior of absorption, distribution, metabolism, and excretion (ADME)

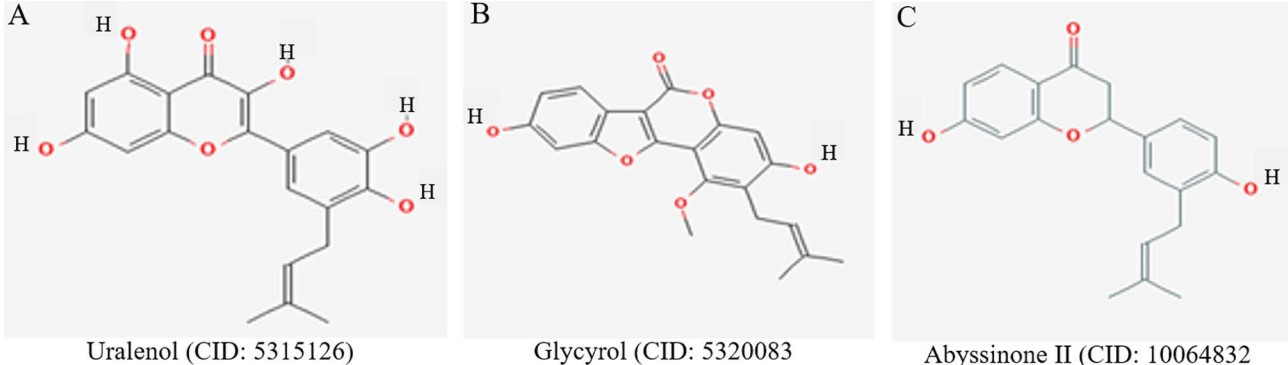

**Fig 2. 2D structure of the lead compounds.** The 2D structure of uralenol, glycyrol, and abyssinone II compounds was selected based on their docking and inhibitory activity. The chemical structures were generated using the 2D Sketcher (Beta) software.

[50]. The selected phytochemicals demonstrated favorable ADME profiles, a critical requirement for small molecules to be considered as potential drug candidates, as summarized in Table 2. Furthermore, all four compounds exhibited low toxicity levels, suggesting their suitability for safe administration. Table 3 displays the toxicity characteristics of the chosen compounds along with the reference control.

## MD simulation

MD simulation serves a vital function in elucidating the dynamic characteristics of biomolecular systems in aqueous environments by efficiently monitoring their motion patterns over different temporal scales and providing information about thermal-averaged molecular attributes that correspond well with experimental ensemble measurements [51]. By calculating bulk solution properties and assessing free energy modifications, MD simulation fulfills a crucial role in clarifying mechanisms like molecular binding interactions [52]. MD simulation was performed on the highest-scoring complexes to assess their structural stability and flexibility throughout a 100 ns timeframe, aiming to identify potentially effective inhibitors. Multiple parameters, including RMSD, RMSF, radius of gyration (Rg), solvent-accessible surface area (SASA), free energy landscape, and principal component analysis (PCA), were analyzed for both the unbound 6LVK structure and its most favorable compound-bound complex.

## RMSD analysis of the complexes

RMSD analysis was employed to assess stability and conformational dynamics of the FGFR2–ligand complexes over a 100 ns MD simulation (Fig 5). Lower RMSD values indicate greater structural stability, with reduced fluctuations reflecting a more stable protein–ligand complex [53]. The RMSD profiles of FGFR2 in its apo form and complex with CID: 5315126, CID: 5320083, CID: 10064832, and the control compound CID: 44243159 were analyzed. The average RMSD values for the apo protein, control, CID: 5315126, CID: 5320083, and CID: 10064832 were 1.72 Å, 1.77 Å, 1.88 Å, 1.21 Å, and 1.33 Å, respectively.

Among these, CID: 5320083 showed the lowest average RMSD (1.21 Å), indicating superior conformational stability and minimal structural deviation. Its RMSD peaked at 2.81 Å around 71.19 ns and stabilized between 0.7–1.5 Å after the initial 0.1 ns. In contrast, the apo FGFR2 began at 0 Å, reached 2.51 Å at 72.36 ns, and stabilized between 0.9–1.2 Å after 0.5 ns, reflecting moderate flexibility. The control compound CID: 44243159 exhibited a peak of 2.59 Å at 13.13 ns and stabilized between 1.0–1.5 Å, ending at 1.79 Å, suggesting consistent structural behavior. CID: 10064832 exhibited moderate conformational stability, with an RMSD peak of 2.83 Å at 53.29 ns and stabilization within the 1.0–1.5 Å range after 0.5 ns, ending at 1.37 Å. CID: 5315126, despite showing the highest average RMSD (1.88 Å) and a peak of 3.07 Å

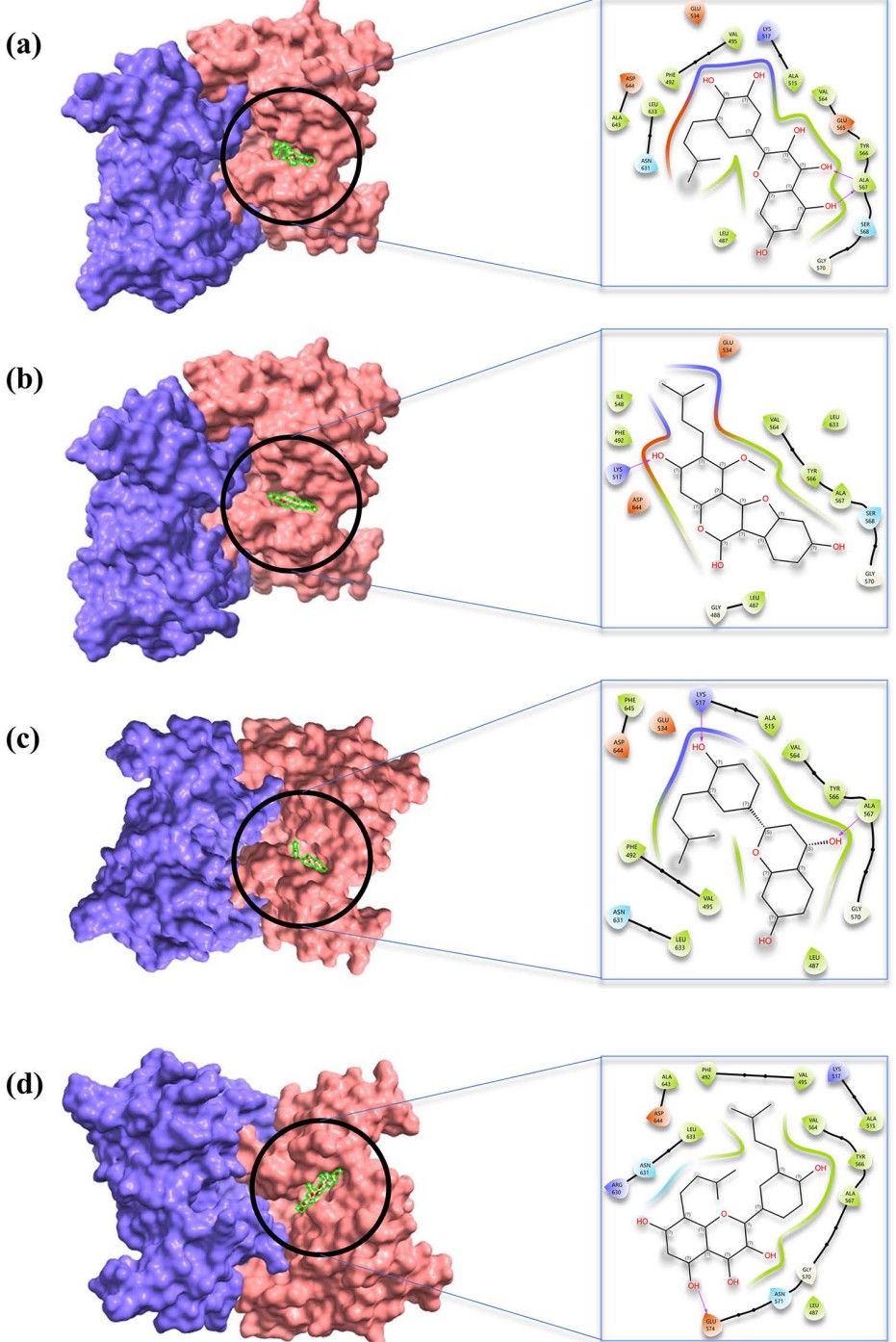

**Fig 3. Interaction diagrams of the lead compounds.** Two-dimensional and three-dimensional interaction diagrams of (A) uralenol, (B) glycyrol, (C) abyssinone II, and (D) control compound in complex with FGFR2.

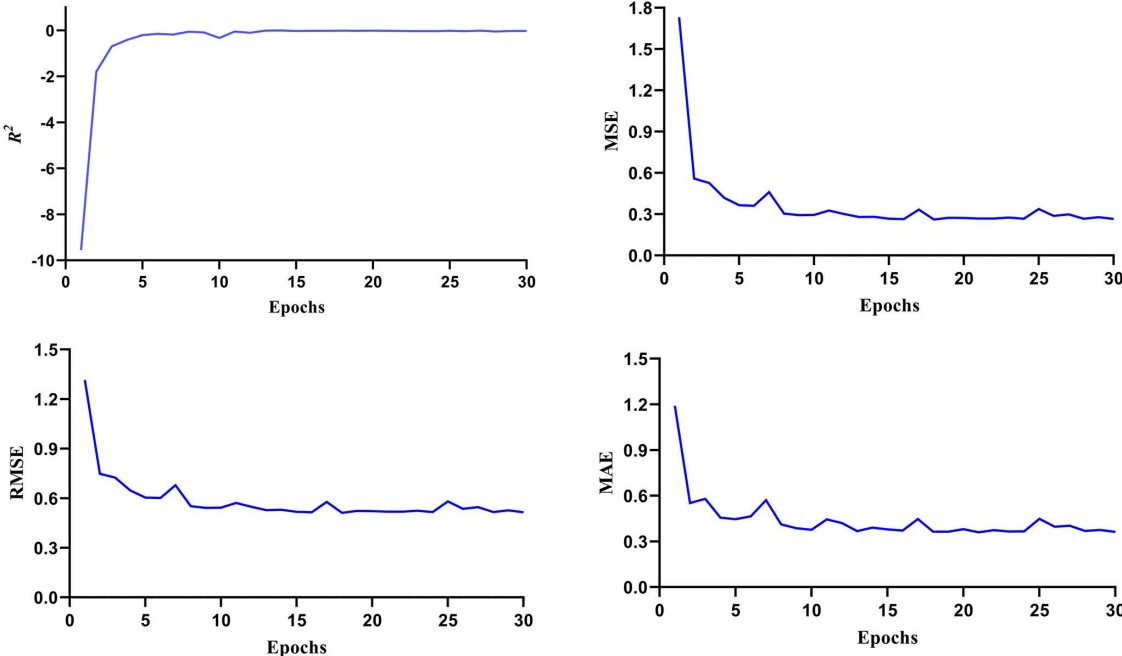

**Fig 4. Analysis of the deep learning model.** Performance of the deep learning regression model used to predict FGFR2 inhibitory activity, including values for $R^2$, MSE, RMSE, and MAE.

**Table 2. ADME and physicochemical properties of selected lead compounds, including molecular weight, TPSA, LogP, solubility, GI absorption, and BBB permeability.**

| Parameters | Properties | CID 5315126 | CID 5320083 | CID 10064832 | CID 44243159 |
|---|---|---|---|---|---|
| **Physiochemical properties** | MW (g/mol) | 370.11 | 366.11 | 324.14 | 494.16 |
| | Heavy atoms | 27 | 36 | 31 | 50 |
| | Aromatic Atoms | 12 | 14 | 12 | 18 |
| | Rotatable bond | 3 | 3 | 3 | 4 |
| | H-bond Acceptors | 7 | 6 | 4 | 9 |
| | H-bond donors | 5 | 2 | 2 | 5 |
| | TPSA (Å$^2$) | 131.36 | 93.04 | 66.76 | 138.07 |
| **Lipophilicity** | Log P$_{o/w}$ (Cons) | 3.171 | 3.84 | 3.97 | 1.256 |
| **Water solubility** | Log S (ESOL) | −4.674 | −5.07 | −4.89 | −5.65 |
| **Pharmacokinetics** | GI absorption | High | High | Moderate-high | High |
| | BBB permeant | No | No | Yes | Yes |
| | P-GP substrate | Yes | Yes | Yes | Yes |
| **Drug likeness** | Lipinski | Yes | Yes | Yes | Yes |
| **Medi. chemistry** | Synth. accessibility | Easy to synthesize | Easy to synthesize | Easy to synthesize | Easy to synthesize |

TPSA; topological polar surface area, which predicts drug transport properties, GI absorption; which predicts gastrointestinal absorption (High/Low), BBB permeability; ability to cross the blood–brain barrier, Log P$_{o/w}$ (Cons); lipophilicity (octanol/water partition coefficient), Log S (ESOL); predicted aqueous solubility, P-GP substrate; indicates if compound is a P-glycoprotein substrate.

**Table 3. Predicted toxicity profiles of selected compounds, including hepatotoxicity, carcinogenicity, mutagenicity, immunotoxicity, LD$_{50}$ values, and overall toxicity class.**

| Target | CID 5315126 | CID 5320083 | CID 10064832 | CID 44243159 |
|---|---|---|---|---|
| **Hepatotoxicity** | Active | Inactive | Inactive | Inactive |
| **Carcinogenicity** | Inactive | Inactive | Inactive | Inactive |
| **Mutagenicity** | Inactive | Active | Inactive | Inactive |
| **Cytotoxicity** | Inactive | Inactive | Inactive | Inactive |
| **LD$_{50}$ (mg/kg)** | 1190 | 500 | 2000 | 1000 |
| **Immunotoxicity** | Active | Active | Active | Active |
| **Toxicity Class** | 4 | 4 | 4 | 4 |

LD$_{50}$: Median lethal dose (mg/kg), predicts acute toxicity.

**Fig 5. Molecular dynamics analysis of the compounds.** Molecular dynamics analysis of FGFR2 in apo form, control, and ligand-bound complexes over 100 ns simulation. The RMSD profiles indicate enhanced structural stability upon ligand binding. RMSF analysis shows reduced residue fluctuations in complexes compared to the apo form. Radius of gyration (Rg) values suggest compact and stable structures, while SASA profiles reveal consistent solvent exposure.

at 82.83 ns, also stabilized consistently between 1.7–2.3 Å after 1.0 ns. Both compounds maintained structural integrity without significant fluctuations, supporting their binding stability.

## RMSF analysis of the complexes

RMSF represents a key metric in MD simulations, employed to assess the mobility of individual amino acid residues within a protein receptor throughout ligand interactions across the simulation period. RMSF quantifies the fluctuation of atomic coordinates relative to their mean positions during the MD trajectory, offering detailed information about residue-level dynamic behavior. In this study, we conducted RMSF analysis on each residue over a 100 ns MD simulation, enabling the identification of regions exhibiting increased flexibility within the protein structure. Notably, residues ASP471, LYS507,

LYS659, and ASN766 exhibited pronounced flexibility during the simulation. These residues are known for their participation in ligand interaction, as demonstrated in Fig 5. Of these residues, LEU468 exhibited the greatest RMSF value, attaining 1.42 Å. Average RMSF values observed for the apo form (FGFR2), control, CID 5315126, CID 5320083, and CID 10064832 were 0.138 Å, 0.125 Å, 0.133 Å, 0.134 Å, and 0.149 Å, respectively. Among the systems, the control compound exhibited the lowest average RMSF, suggesting relatively stable binding. However, both CID 5315126 and CID 5320083 showed comparable minor fluctuations, indicating that these ligands stabilize the protein structure to a similar extent. Notably, CID: 10064832 displayed the highest average RMSF, implying increased local flexibility in certain regions of the protein, which may be attributed to conformational adaptations upon ligand binding.

## RG analysis of the complexes

RG measures the spatial arrangement of atoms relative to the central axis in a protein-ligand complex throughout a defined simulation timeframe. Calculating the Rg is crucial for assessing the structural stability of a macromolecule, as it provides important information regarding alterations in the complex's compactness. In this study, the Rg of FGFR2 was assessed over a 100 ns MD simulation for the apo form, the control compound (CID 44243159), and three ligand-bound complexes (CID: 5315126, CID: 5320083, and CID: 10064832). The average Rg values recorded were 1.984 Å for the apo protein, 1.995 Å for the control, 1.999 Å for CID: 5315126, 1.999 Å for CID: 5320083, and 2.005 Å for CID: 10064832, as depicted in Fig 5. The Rg values across all complexes were closely aligned, without any major structural shift in the active site of the protein after binding. Notably, the apo and control systems displayed greater fluctuations in Rg throughout the 100 ns simulation period, suggesting a relatively higher degree of structural flexibility. In contrast, the three ligand-bound complexes demonstrated more consistent Rg profiles, implying enhanced conformational stability and maintenance of protein compactness upon ligand binding. These findings highlight the stabilizing effect of ligand interaction on FGFR2 structure, which maintained compact and stable conformations throughout the simulation.

## SASA analysis of the complexes

Solvent Accessible Surface Area (SASA) measures the fraction of a protein's surface that remains available for interaction with solvent molecules [54]. It provides valuable molecular-level insights into conformational fluctuations and the nature of ligand interactions with protein macromolecules, as illustrated in Fig 5. The analysis was conducted over a 100 ns MD simulation for the apo FGFR2 protein, the control complex (CID: 44243159), and three test compound complexes with 6LVK proteins (CID: 5315126, CID: 5320083, and CID: 10064832). Average SASA values were 1009.677 Å² for the apo protein, 1010.493 Å² for the control, 1009.981 Å² for CID: 5315126, 1010.075 Å² for CID: 5320083, and 1009.692 Å² for CID: 10064832. These values are very close across all systems, indicating a consistent solvent exposure profile during the simulation. In terms of fluctuation ranges, the apo system exhibited SASA values between 996.58 and 1018.97 Å², reflecting inherent flexibility in the unbound state. The control complex showed a slightly wider range, from 996.78 to 1021.23 Å², suggesting minor structural rearrangements upon ligand binding. In comparison, all lead phytochemicals— CID: 5315126, CID: 5320083, and CID: 10064832—in complex with FGFR2 exhibited relatively stable SASA trajectories, characterized by minimal fluctuations throughout the 100 ns simulation period.

## Principal component analysis

PCA analysis revealed distinct sampling behaviors among the five systems (A–E). Apo A and Control B exhibited extensive conformational exploration, with broad distributions along PC1 (−125–75) and PC2 (−100–50). Complexes D and E showed more constrained sampling, with tight clustering in the PCA space, while Complex C displayed intermediate dispersion. Quantitatively, the standard deviation of PC1 was ~40–50 for A and B, and ~25–30 for D and E (Fig 6).

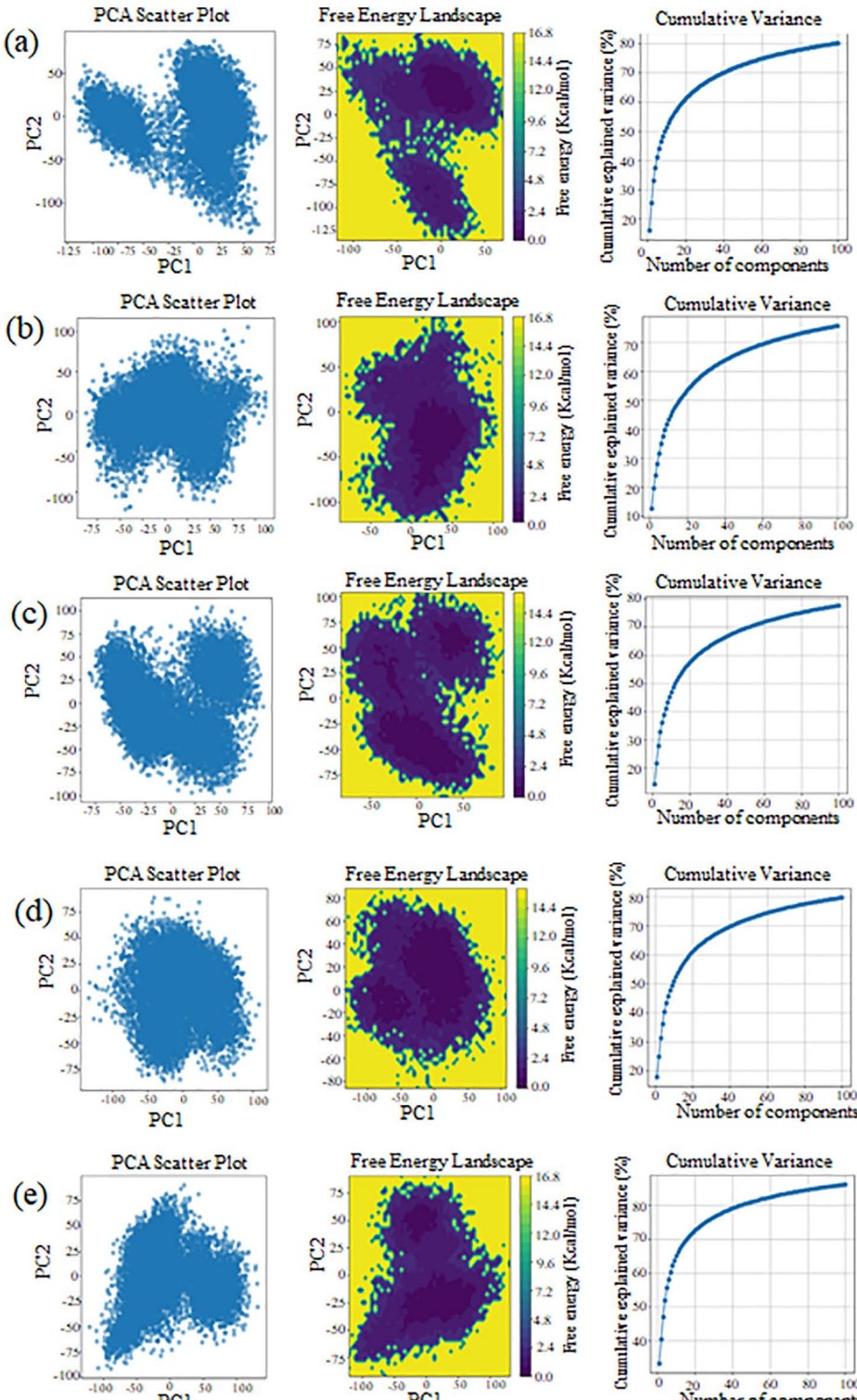

**Fig 6. Principal component analysis (PCA) of the compounds.** PCA scatter plots, free energy landscapes, and cumulative variance plots for molecular dynamics simulations of: (A) apo form, (B) control system, (C) uralenol complex, (D) glycyrol complex, and (E) abyssinone II complex. PCA reveals conformational sampling patterns, energy landscapes show stable states (purple=low energy, yellow=high energy), and variance plots indicate dimensional contribution to system dynamics.

## Free energy landscapes

Free energy surfaces varied across systems. Apo A and Control B showed complex landscapes with multiple shallow basins and energy values ranging from 0.0 to 16.8 kJ/mol. Complex D presented a deep, centralized energy minimum surrounded by steep energy gradients (>10 kJ/mol). Complexes C and E showed intermediate features: Complex C had a broader low-energy basin, while Complex E displayed an asymmetric energy surface. However, cumulative variance plots were similar across systems. The first 20 principal components captured 50–60% of total variance, while 40–60 components explained up to 80%, indicating comparable dimensionality for essential dynamics across all systems.

## Discussion

FGFR2 is a transmembrane receptor tyrosine kinase that plays a critical role in regulating essential cellular processes, including proliferation, differentiation, and cell survival [16]. Accurate control of FGFR2 signaling is crucial for maintaining proper cellular activities and tissue balance. Disruptions in FGFR2 signaling, caused by genetic mutations or abnormal expression levels, have been linked to the onset of numerous diseases, especially malignancies [55]. Acknowledged as an oncogenic driver, FGFR2 has become an attractive target for therapeutic exploration. Selectively inhibiting its aberrant activity offers the potential to suppress cancer cell proliferation and survival while minimizing adverse effects on normal cells [56]. The development of novel FGFR2 inhibitors has attracted considerable attention in the field of precision medicine, as these agents may enable personalized treatment strategies based on patients' specific genetic alterations and signaling pathway dysregulations. Such targeted therapies could significantly improve clinical outcomes by directly targeting the molecular drivers of disease, potentially overcoming current therapeutic limitations in cancer treatment [57]. The Fibroblast Growth Factor (FGF) family consists of a collection of signaling proteins that function through FGFRs, controlling numerous biological processes including embryonic growth, tissue regeneration, and blood vessel formation. Within the FGFR variants (FGFR1 through FGFR4), FGFR2 stands out due to its widespread tissue expression and its essential role in developmental mechanisms and physiological balance maintenance [57,58]. Disrupted FGFR2 function, caused by gene amplifications, mutations, or excessive expression, can result in overactive signaling pathways that facilitate tumor initiation and cancer advancement. Significantly, FGFR2 alterations have been reported in numerous cancer types, emphasizing its clinical significance as a promising therapeutic target [59,60]. Natural products have long been recognized as valuable sources of therapeutic agents for a wide range of human diseases and have been utilized in medicine since ancient times [61].

Virtual screening and computer-aided drug design have become indispensable approaches in modern drug discovery, leveraging advanced computational methodologies to identify and develop novel therapeutic compounds [62]. These methods have thus become indispensable in CADD, particularly for the virtual screening of natural bioactive phytochemical libraries to identify potential drug candidates [63]. Molecular docking represents a core technique in pharmaceutical research and development, facilitating the prediction of binding interactions between small molecules and their target proteins. The binding capacity of the compounds with FGFR2 was determined and prioritized according to AutoDock Vina scoring results. The most favorable affinity from multiple conformational poses of each compound was chosen, as it indicates the maximum energy release upon complex formation [49]. Molecular docking evaluation revealed sixteen compounds displaying binding energies greater than −9.0 kcal mol⁻¹, demonstrating robust interactions with the FGFR2 protein, as presented in Table 1. These findings are consistent with the recognized concept that more negative binding energies generally correlate with enhanced stability and effectiveness of ligand-target associations [64]. A significant characteristic of the binding interactions observed in this investigation is the participation of hydrogen bonding, hydrophobic interactions, and additional non-covalent forces (Fig 3). These molecular interactions are presumably vital for strengthening the structural integrity of the protein-ligand complexes and may enhance their inhibitory efficacy against the target protein.

In recent years, deep learning approaches have emerged as powerful tools in drug discovery, enabling the efficient prediction of molecular bioactivity from large chemical datasets. The results demonstrate that deep learning-based regression modeling can effectively predict compound bioactivity against viral targets, as shown by the high R² value and low error metrics. The model's performance reflects its ability to learn complex structure-activity relationships from topological and chemical features encoded in the fingerprint data. The predicted $pIC_{50}$ values of the selected compounds closely matched experimental expectations, validating the model's screening capability. Among them, CID 5315126, CID 5320083, and CID 10064832 exhibited pIC50 values of 5.21, 5.06, and 4.66, respectively, indicating that these compounds possess notable biological activity and potential inhibitory effects on the target protein. Our deep learning predictions are constrained by reliance on available datasets and descriptor selection, highlighting the need for larger, diverse training sets and experimental validation for broader applicability.

The ADME analysis presented in Table 2 demonstrates that uralenol, glycyrol, and abyssinone II exhibit favorable pharmacokinetic properties, including optimal molecular weight, LogP values, and topological polar surface area (TPSA), suggesting suitable membrane permeability and oral bioavailability. All compounds showed high gastrointestinal (GI) absorption, acceptable water solubility, and no predicted blood-brain barrier (BBB) penetration, reducing the risk of central nervous system-related side effects. These characteristics enhance their drug-likeness and systemic applicability. In Table 3, toxicity profiling further supports the therapeutic potential of these candidates. None of the compounds showed predicted hepatotoxicity, carcinogenicity, or mutagenicity. Their predicted $LD_{50}$ values fall within acceptable safety margins, placing them in lower toxicity classes. The absence of major immunotoxic or cytotoxic alerts indicates a lower risk of adverse effects. Collectively, the ADME and toxicity results suggest that these phytochemicals are not only bioactive but also possess drug-like safety and absorption profiles suitable for further preclinical development as FGFR2 inhibitors.

MD simulation is utilized to evaluate protein stability following ligand binding. It additionally offers information regarding the structural integrity and flexibility of protein-ligand complexes across a specified time period, replicating physiological environments comparable to those found in the human body [65]. The MD simulations performed in this research provided a valuable understanding of the structural stability and dynamic characteristics of FGFR2 protein-ligand complexes throughout a 100 ns duration. Analysis of essential parameters such as RMSD, RMSF, RG, SASA, FEL, and PCA contributed to clarifying the structural features and general behavior of these molecular complexes. RMSD calculations provide an assessment of the overall stability of protein–ligand complexes, while RMSF values offer insights into the fluctuations of individual residues during the ligand-binding process [65]. During the 100 ns simulation timeframe, the three selected compounds, CID 5315126, CID 5320083, and CID 10064832, demonstrated reduced RMSD and RMSF values upon receptor binding, suggesting improved structural integrity of the complexes, as shown in Fig 5. A decreased radius of gyration (Rg) value reflects increased structural compactness, while elevated values imply a propensity for the compounds to separate from the protein [66]. In comparison to the CID 44243159 protein complex and the unbound protein, the complexes generated by the three leading compounds displayed consistent Rg values with minimal variations. Reduced SASA values signify more condensed complexes featuring stronger interactions between amino acid residues and adjacent water molecules, while increased values indicate diminished structural integrity. In this investigation, the complexes of CID 5315126, CID 5320083, and CID 10064832 with the receptor demonstrated reduced SASA values relative to the protein-CID 44243159 complex, as depicted in Fig 5.

The PCA and free energy analyses highlight differences in molecular flexibility and stability. Apo A and Control B extensive sampling and multiple energy basins suggest high conformational flexibility and the presence of metastable states (Fig 6). Such behavior is often associated with molecules requiring dynamic structural shifts, such as allosteric proteins. In contrast, complex D narrow sampling range and a deep energy minimum, indicating structural rigidity and thermodynamic stability, likely maintained by strong intramolecular interactions. This is typical of proteins or ligands with a well-defined functional conformation. Complexes C and E represent intermediate behaviors. Complex C broad energy basin suggests a flexible yet stable structure, potentially allowing moderate conformational shifts. Complex E asymmetric energy profile

may reflect directional preferences or internal constraints affecting dynamics. Similar cumulative variance curves validate PCA's effectiveness for dimensionality reduction and underscore its utility in conformational analysis. Combining PCA with free energy mapping provides a comprehensive picture of molecular behavior, offering insights into both the extent and thermodynamic feasibility of conformational states. Recent studies report several FGFR inhibitors, including AZD4547, BGJ398, dovitinib, and lucitanib, progressing to phase II trials, though none are approved for breast cancer [67,68]. These synthetic inhibitors face challenges of selectivity, resistance, and safety [69,70]. In contrast, our computational analysis identified natural compounds, uralenol, glycyrol, and abyssinone II, with favorable binding, stability, and pharmacokinetics. Thus, our findings suggest phytochemicals may provide safer and more effective alternatives for FGFR2-targeted therapy.

Recent efforts to target FGFR2 signaling in cancer have largely focused on synthetic small-molecule tyrosine kinase inhibitors, such as AZD4547, BGJ398 (infigratinib), and pemigatinib, which exhibit high potency but are often associated with dose-limiting toxicities, acquired resistance, and suboptimal pharmacokinetic profiles [71–74]. In this context, phytochemical-based inhibitors have gained increasing attention due to their structural diversity, favorable safety profiles, and potential for multitarget modulation. Our integrative in-silico approach combining molecular docking and molecular dynamics simulations provides a comparative framework to evaluate the FGFR2 inhibitory potential of natural compounds relative to previously reported inhibitors. Very recent studies on FGFR2 inhibitors reported good binding affinities of −8.0 to −9.6 kcal/mol for synthetic compounds [75], and −8.3801 kcal/mol, −8.3014 kcal/mol, and −8.320 kcal/mol for Knipholone, Torosanin, and Rubrolide E, respectively [74]. Our phytochemicals, uralenol (−9.8 kcal/mol), glycyrol (−9.7 kcal/mol), and abyssinone II (−9.7 kcal/mol), showed strong binding affinity towards the FGFR2 kinase domain, comparable to or exceeding those reported in earlier studies. Clinical FGFR inhibitors like erdafitinib and infigratinib exhibit moderate structural stability in MD simulations [67,68], whereas our compounds demonstrated superior stability (RMSD: 1.21–1.88 Å). Previous phytochemical studies targeting FGFRs showed weaker activity [73]. Unlike synthetic inhibitors facing selectivity and resistance challenges [69], with low toxicity predictions, our compounds exhibited favorable ADME properties, reduced RMSD fluctuations, favorable radius of gyration profiles, and stable hydrogen bond occupancy throughout the simulation period. Overall, when contextualized against existing FGFR2 inhibitors and phytochemical-based kinase modulators, our findings suggest that uralenol, glycyrol, and abyssinone II represent promising scaffolds with balanced binding affinity, structural stability, pharmacokinetic characteristics, and superior safety profiles, suggesting their potential relevance as lead candidates.

This study is entirely computational and lacks experimental validation. While molecular docking, deep learning, and molecular dynamics simulations provide valuable predictive insights, in vitro and in vivo studies are essential to confirm biological efficacy and therapeutic potential. The ADME/T and toxicity profiles were predicted using in silico models, which may not fully reflect complex human physiological responses. The focus on a subset of top-ranked compounds may have excluded other potential candidates. Therefore, our findings should be regarded as predictive rather than confirmatory. Expanding the chemical space and conducting experimental validation, such as in vitro kinase inhibition assays, cell viability tests in FGFR2-overexpressing cancer cell lines, and in vivo xenograft models, are essential next steps to validate these computational predictions and advance these compounds toward preclinical development.

## Conclusion

This study successfully identified uralenol, glycyrol, and abyssinone II as potent and stable FGFR2 inhibitors from a phytochemical library of traditionally used anti-cancer plants using an integrated in silico framework. The integrated approach of molecular docking, deep learning prediction, ADME/T evaluation, and molecular dynamics simulations provided strong evidence for the structural stability, binding affinity, bioactivity, and favorable pharmacokinetic properties of these compounds toward FGFR2. Competitive binding affinities (−9.7 to −9.8 kcal/mol) comparable to synthetic inhibitors, superior structural stability (RMSD: 1.21–1.88 Å) during MD simulations, favorable drug-like properties with minimal predicted toxicity, and stable conformational dynamics confirmed by PCA and free energy landscape analyses further confirmed

stable and energetically favorable protein–ligand interactions. Importantly, this work highlights the novel potential of these phytochemicals as FGFR2-targeted anticancer candidates, expanding the scope of plant-derived kinase inhibitors with enhanced stability and safety profiles compared to current synthetic therapeutics, addressing limitations of off-target toxicity and resistance. These computational predictive findings support the therapeutic potential of the compounds and highlight them as potential candidates for further validation through *in vitro* kinase inhibition assays against FGFR2, cell viability and proliferation studies in FGFR2-overexpressing cancer cell lines, and *in vivo* xenograft models to assess antitumor efficacy. Successful experimental validation could position these natural compounds as safer alternatives for FGFR2-targeted cancer therapy.

## Supporting information

**S1 Table. List of anti-cancer properties of medicinal plants with their 1350 Phytochemicals.**
(DOCX)

## Acknowledgments

We would like to express our sincere gratitude to Abdul Kuddus, Sajib Ghosh, Md. Borhan Uddin, Md. Akon Bin Kawsar, and Sumaiya Tasnime for their valuable support and contributions to the literature review process. Their efforts and dedication greatly enhanced the quality and depth of this research.

## Author contributions

**Conceptualization:** Alomgir Hossain, Mohammad Nurul Matin.

**Data curation:** Alomgir Hossain, Md. Sanowar Hossan, Muntasir Rahman Siam, Md. Ekhtiar Rahman.

**Formal analysis:** Alomgir Hossain, Md. Shahanur Prodhan, Md. Nahid Hasan Joy.

**Funding acquisition:** Mohammad Nurul Matin.

**Investigation:** Md. Sanowar Hossan.

**Methodology:** Md. Ekhtiar Rahman.

**Software:** Alomgir Hossain, Md. Shahanur Prodhan, Md. Nahid Hasan Joy, Md. Ekhtiar Rahman.

**Supervision:** Mohammad Nurul Matin.

**Writing – original draft:** Alomgir Hossain.

**Writing – review & editing:** Md. Sanowar Hossan, Md. Shahanur Prodhan, Md. Nahid Hasan Joy, Muntasir Rahman Siam, Md. Ekhtiar Rahman.

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
