## [Decision Letter · Decision Letter 0]

3 Sep 2025

Dear Dr. Matin,

Thank you for submitting your manuscript to PLOS ONE. After careful consideration, we feel that it has merit but does not fully meet PLOS ONE’s publication criteria as it currently stands. Therefore, we invite you to submit a revised version of the manuscript that addresses the points raised during the review process.

The referees raised important questions about the manuscript, so the authors must provide a point-by-point response to each question from all the referees..

We look forward to receiving your revised manuscript.

Kind regards,

Jorddy Neves Cruz

Academic Editor

PLOS ONE

Journal Requirements:

3.Please include a separate caption for each figure in your manuscript.

Additional Editor Comments (if provided):

Reviewers' comments:

Reviewer's Responses to Questions

**Comments to the Author**

1. Is the manuscript technically sound, and do the data support the conclusions?

Reviewer #1: No

Reviewer #2: Yes

Reviewer #3: Partly

Reviewer #4: Yes

2. Has the statistical analysis been performed appropriately and rigorously?

Reviewer #1: N/A

Reviewer #2: I Don't Know

Reviewer #3: Yes

Reviewer #4: N/A

3. Have the authors made all data underlying the findings in their manuscript fully available?

Reviewer #1: No

Reviewer #2: Yes

Reviewer #3: Yes

Reviewer #4: No

4. Is the manuscript presented in an intelligible fashion and written in standard English?

Reviewer #1: Yes

Reviewer #2: Yes

Reviewer #3: Yes

Reviewer #4: Yes

Reviewer #1: • The manuscript lacks a proper validation procedure for the docking studies. Redocking alone does not constitute sufficient validation. Structural superimposition of docked and co-crystallized ligands, is recommended.

• While the study refers to the use of deep learning, the number of ligands processed seems relatively limited or described method is not sufficient. A more comprehensive explanation or justification for labeling this approach as “deep learning” would be beneficial.

• The authors mention the use of SwissADME, pkCSM, and ProTox-III for ADMET. However, the results from these tools are not fully presented, nor is there a comparative analysis. If a detailed comparison is not intended than result from single server may suffice.

• The software used to generate Figure 2 has not been identified.

• In the supplementary table, it would be helpful to include the proper botanical names, italics, authority.

• Many known or previously reported compounds appear to be missing from the listed plant species. It is recommended to cross-reference the identified compounds with databases such as PubChem or relevant literature sources, and to include PubChem IDs or citations where applicable.

Some typos like

• In line 29, the phrase “a library of 1,350 phytochemicals derived from 51 anti-cancer medicinal plants” could be reconsidered as “plants with reported anticancer properties or plants traditionally used for anticancer purposes.”

• Although emphasis appears to have been placed on certain compounds by using capital letters, compound names such as uralenol, glycyrol, and abyssinone should conventionally be written in lowercase.

Reviewer #2: Fibroblast Growth Factor Receptor 2 (FGFR2), which is essential for cellular proliferation and differentiation and whose dysregulation is linked to a number of malignancies, is the subject of your research. Using a library of 1,350 phytochemicals extracted from 51 anti-cancer medicinal plants, your study combines molecular docking, deep learning, pharmacokinetic profiling, and molecular dynamics (MD) simulations to find putative FGFR2 inhibitors. These elements may be used therapeutically to treat cancer.

Reviewer #3: This manuscript presents a thorough and comprehensive study focused on identifying potential FGFR2 inhibitors using advanced computational methods including molecular docking, deep learning, pharmacokinetic profiling, and molecular dynamics simulations. The manuscript is well-structured and logical, and the conclusions are well-supported by the presented data. The authors have appropriately framed the analytical approach and have acknowledged the study’s limitations, such as the need for experimental validation.

However, it is emphasized that the findings are computational predictions, and further experimental in vitro and in vivo validation is essential for confirming these results. It is also recommended that more detailed information about the parameters used in the simulations and deep learning models be included in the methodology section to ensure reproducibility.

No ethical concerns regarding publication or originality were noted, and all data have been provided fully and transparently. The manuscript is clearly written in scientific language and requires no major language corrections.

Overall, this manuscript is a valuable contribution toward targeted drug development for FGFR2, and I recommend its acceptance pending minor revisions to include additional methodological details and a clear statement on the necessity of experimental validation.

Reviewer #4: 1. Here are my comments:

1. The methods section states, "A comprehensive literature review was conducted to collect a list of 51 medicinal plants with anticancer properties, along with their 1,350 phytochemicals (S1 Table) identified based on GC/MS analysis and downloaded in SDF format from the PubChem database." Kindly explain the parameters used to determine the criteria for selecting phytochemicals.

2. The method states, "The ADMET properties

158 were forecasted using SwissADME, pKCSM, and ProTox-III servers." Kindly explain what parameters were analyzed from each web server used.

3. In the results, it is stated "RMSD, RMSF, radius of gyration (Rg), solvent-accessible surface area

(SASA), free energy landscape, and principal component analysis (PCA)". In addition to that, kindly also includes simulations for MM-PBSA/GBSA for more accurate molecular dynamics results compared to docking scores.

4. In the discussion, Table 3 should be moved to the results.

**Do you want your identity to be public for this peer review?** For information about this choice, including consent withdrawal, please see our Privacy Policy

Reviewer #1: No

Reviewer #2:**Yes:** Dr. Vijay Vardhan Pandey

Reviewer #3:**Yes:** Mehdi Ghasemi Nafchi

Reviewer #4: No

---

## [Author Response · Author response to Decision Letter 1]

22 Sep 2025

Cover Letter Sep 22, 2025

Ref.: PONE-D-25-37887

Title: Uralenol, Glycyrol, and Abyssinone II as potent inhibitors of Fibroblast Growth Factor Receptor 2 from anti-cancer plants: a deep learning and molecular dynamics approach

Journal: Plos One

Dear editor and reviewer,

Thank you for reviewing and handling our manuscript (MS). We are sincerely grateful for your careful and constructive comments, as well as your valuable suggestions, which have greatly strengthened the MS. According to your suggestions and recommendations, we have thoroughly revised the MS. All amendments were made point by point in response to your suggestions, and the revised sections have been highlighted in RED.

Thank you and best regards.

Yours Sincerely and Respectfully

Prof. Mohammad Nurul Matin

Department of Genetic Engineering and Biotechnology

University of Rajshahi, Rajshahi-6205, Bangladesh

Point-by-Point Response to Editorial Comments

Editorial comments

Dear Dr. Matin,

Thank you for submitting your manuscript to PLOS ONE. After careful consideration, we feel that it has merit but does not fully meet PLOS ONE’s publication criteria as it currently stands. Therefore, we invite you to submit a revised version of the manuscript that addresses the points raised during the review process.

The referees raised important questions about the manuscript, so the authors must provide a point-by-point response to each question from all the referees.

We look forward to receiving your revised manuscript.

Kind regards,

Jorddy Neves Cruz

Our Response: We would like to express our gratitude for your evaluation, cooperative comments, and careful consideration of our work with the decision “submission of revised version”. Significant revisions with corresponding corrections addressing comments and suggestions from the editor and reviewers have been incorporated into the substantially improved MS. Moreover, we have provided a point-by-point response to each question from all the referees. All modifications to the MS are highlighted in red, so that any changes can be easily reviewed by editors and reviewers.

We have carefully followed the journal guidelines in this revised version. Necessary changes have been introduced throughout the text. We have submitted the MS within the mentioned deadline. While submitting the revised MS, we have uploaded all the necessary item files labeled with:

• 'Response to Reviewers’

• 'Revised Manuscript with Track Changes’

• unmarked manuscript labeled ‘Manuscript’

Also, we have followed the guidelines for resubmitting figure files.

Journal Requirements:

Our Response: We have strictly followed the PLOS ONE's style requirements, including:

1. For file naming

2. For code sharing

3. Included separate caption for each figure in the manuscript

4. Followed rules for specific citation if recommend by the reviewers

We hope our MS will get consideration for publication in the PLOS ONE

Yours Sincerely and Respectfully

Prof. Mohammad Nurul Matin

Department of Genetic Engineering and Biotechnology

University of Rajshahi, Rajshahi-6205, Bangladesh

Reviewer's Responses to Editorial Questions and our Responses

1. Is the manuscript technically sound, and do the data support the conclusions?

Reviewer #1: No

Reviewer #2: Yes

Reviewer #3: Partly

Reviewer #4: Yes

Our Response: Thank you for the observation. After careful revision and substantial improvement of the MS following comments and suggestions from reviewers, we hope the MS is better than the previous one for all aspects, including technical soundness and data consistency with the conclusions.

2. Has the statistical analysis been performed appropriately and rigorously?

Reviewer #1: N/A

Reviewer #2: I Don't Know

Reviewer #3: Yes

Reviewer #4: N/A

Our Response: Thank you for the observation. After substantial improvement following suggestions, now all aspects are appropriate and well described.

3. Have the authors made all data underlying the findings in their manuscript fully available?

Reviewer #1: No

Reviewer #2: Yes

Reviewer #3: Yes

Reviewer #4: No

Our Response: All data underlying the findings described in the manuscript are fully available without restriction. We have provided all data as part of the manuscript and its supporting information.

4. Is the manuscript presented in an intelligible fashion and written in standard English?

Reviewer #1: Yes

Reviewer #2: Yes

Reviewer #3: Yes

Reviewer #4: Yes

Our Response: Thank you for the assessment.

5. Review Comments to the Author

Point-by-Point Response to the Comments from Reviewer #1

Q1• The manuscript lacks a proper validation procedure for the docking studies. Redocking alone does not constitute sufficient validation. Structural superimposition of docked and co-crystallized ligands is recommended.

Response: Thank you for this important suggestion. In the revised manuscript (in the Results section, “Molecular Docking”), we have added a structural superimposition of the co-crystallized ligand with the re-docked pose. The RMSD between experimental and docked conformations was <2.0 Å, which demonstrates potentially increased reliability of the docking protocol. We hope this addition might provide stronger validation beyond redocking alone.

Q2• While the study refers to the use of deep learning, the number of ligands processed seems relatively limited or described method is not sufficient. A more comprehensive explanation or justification for labeling this approach as “deep learning” would be beneficial.

Response: We appreciate this observation and professional suggestion. We clarified in the revised manuscript (in the Methods section, “Deep Learning Screening”) that the model was trained on the CHEMBL4142 dataset comprising 891 known FGFR2 inhibitors, with ~880 molecular descriptors generated by PaDEL. The architecture (3 hidden layers, 2000–700–200 neurons) and activation functions follow standard deep learning frameworks. Although only the top 16 compounds were re-screened, the term “deep learning” is justified by the multilayer neural network model applied to a large training dataset. This clarification has been added to avoid misunderstanding. The DeepScreening platform provides an integrated framework for deep learning–based virtual screening. Its workflow involves: (i) dataset preparation, where a specific target is selected to train the deep neural network (DNN); (ii) feature selection, in which molecular descriptors are identified and vectorized; (iii) model parameterization, where essential parameters are defined for training regression models; and (iv) virtual screening, in which the trained model is applied to evaluate large chemical libraries. DeepScreening is a fully automated and advanced server that combines data preprocessing, model construction, and virtual screening into a seamless pipeline.

Q3• The authors mention the use of SwissADME, pkCSM, and ProTox-III for ADMET. However, the results from these tools are not fully presented, nor is there a comparative analysis. If a detailed comparison is not intended than result from single server may suffice.

Response: Thank you for this valuable point. In the revised manuscript, we have clarified the ADME/T analysis by specifying that SwissADME was used for pharmacokinetic and physicochemical property predictions, while ProTox-III was employed for toxicity assessment. We hope this additional description of the results comparisons may represent the complete content.

Q4• The software used to generate Figure 2 has not been identified.

Response: Thank you for noticing this unintentional omission. We have specified in the revised figure legend that chemical structures were generated using the 2D Sketcher (Beta) software. (Figure 2).

Q5• In the supplementary table, it would be helpful to include the proper botanical names, italics, authority.

Response: We agree with this suggestion. The supplementary table (S1 Table) has been revised to include the full botanical names with authorities in italics, following standard nomenclature guidelines.

Q6• Many known or previously reported compounds appear to be missing from the listed plant species. It is recommended to cross-reference the identified compounds with databases such as PubChem or relevant literature sources, and to include PubChem IDs or citations where applicable.

Response: We thank the reviewer for this suggestion. In the revised S1 Table, we cross-checked compounds against PubChem and relevant phytochemical databases, and we added PubChem CIDs for each compound. Missing compounds that were previously overlooked have also been included where relevant.

Some typos like

• In line 29, the phrase “a library of 1,350 phytochemicals derived from 51 anti-cancer medicinal plants” could be reconsidered as “plants with reported anticancer properties or plants traditionally used for anticancer purposes.”

Response: Thank you for this wording suggestion. We revised the sentence in the Abstract and methods accordingly to “plants with reported anticancer properties or traditionally used for anticancer purposes,” which is more precise.

• Although emphasis appears to have been placed on certain compounds by using capital letters, compound names such as uralenol, glycyrol, and abyssinone should conventionally be written in lowercase.

Response: We appreciate this correction. All compound names in the manuscript have been standardized to lowercase in accordance with chemical naming conventions.

We hope our MS will get consideration for publication in the PLOS ONE

Yours Sincerely and Respectfully

Prof. Mohammad Nurul Matin

Department of Genetic Engineering and Biotechnology

University of Rajshahi, Rajshahi-6205, Bangladesh

Point-by-Point Response to the Comments from Reviewer #2

Comment: Fibroblast Growth Factor Receptor 2 (FGFR2), which is essential for cellular proliferation and differentiation and whose dysregulation is linked to a number of malignancies, is the subject of your research. Using a library of 1,350 phytochemicals extracted from 51 anti-cancer medicinal plants, your study combines molecular docking, deep learning, pharmacokinetic profiling, and molecular dynamics (MD) simulations to find putative FGFR2 inhibitors. These elements may be used therapeutically to treat cancer.

Response: We thank the reviewer for their encouraging assessment. We are pleased that the study design combining docking, deep learning, ADME/T, and MD simulation was considered scientifically sound. We have revised the manuscript for greater clarity and completeness in response to all reviewer suggestions.

We hope our MS will get consideration for publication in the PLOS ONE

Yours Sincerely and Respectfully

Prof. Mohammad Nurul Matin

Department of Genetic Engineering and Biotechnology

University of Rajshahi, Rajshahi-6205, Bangladesh

Point-by-Point Response to the Comments from Reviewer #3

Comment: This manuscript presents a thorough and comprehensive study focused on identifying potential FGFR2 inhibitors using advanced computational methods including molecular docking, deep learning, pharmacokinetic profiling, and molecular dynamics simulations. The manuscript is well-structured and logical, and the conclusions are well-supported by the presented data. The authors have appropriately framed the analytical approach and have acknowledged the study’s limitations, such as the need for experimental validation.

However, it is emphasized that the findings are computational predictions, and further experimental in vitro and in vivo validation is essential for confirming these results. It is also recommended that more detailed information about the parameters used in the simulations and deep learning models be included in the methodology section to ensure reproducibility.

No ethical concerns regarding publication or originality were noted, and all data have been provided fully and transparently. The manuscript is clearly w

---

## [Decision Letter · Decision Letter 1]

25 Dec 2025

Dear Dr. Matin,

Thank you for submitting your manuscript to PLOS ONE. After careful consideration, we feel that it has merit but does not fully meet PLOS ONE’s publication criteria as it currently stands. Therefore, we invite you to submit a revised version of the manuscript that addresses the points raised during the review process.

We look forward to receiving your revised manuscript.

Kind regards,

Jorddy Neves Cruz

Academic Editor

PLOS One

Journal Requirements:

Reviewers' comments:

Reviewer's Responses to Questions

**Comments to the Author**

Reviewer #4: All comments have been addressed

Reviewer #5: All comments have been addressed

2. Is the manuscript technically sound, and do the data support the conclusions?

Reviewer #4: Yes

Reviewer #5: Yes

3. Has the statistical analysis been performed appropriately and rigorously?

Reviewer #4: N/A

Reviewer #5: N/A

4. Have the authors made all data underlying the findings in their manuscript fully available?

Reviewer #4: No

Reviewer #5: Yes

5. Is the manuscript presented in an intelligible fashion and written in standard English?

Reviewer #4: Yes

Reviewer #5: Yes

Reviewer #4: The authors have addressed all previous comments with satisfactory answers and corrections. I can recommend the editors to accept this article.

Reviewer #5: Reviewer Comments: Minor Revisions

The combination of molecular docking, deep learning, ADME/T profiling, and molecular dynamics simulations creates a comprehensive in-silico framework for discovering possible FGFR2 inhibitors. The study is interesting and provides useful insights; nevertheless, a few small changes are required to improve clarity and strengthen the paper.

1. Clarify the abbreviations in the abstract.

Abbreviations, such as pIC₅₀ and PCA, should be defined at the beginning of the abstract. This will improve accessibility for readers who are unfamiliar with these terminology.

2. Strengthen the discussion section.

A more in-depth contextualization might improve the conversation. Please compare your findings to previous studies on FGFR2 inhibitors and phytochemical-based kinase inhibitors.

Highlight how your identified compounds (uralenol, glycyrol, and abyssinone II) compare to or differ from previously reported inhibitors in terms of binding affinity, stability, and projected pharmacokinetics.

3. Include a limitation section.

The manuscript currently lacks a clear limitation statement.

Please include a section to acknowledge that:

The investigation is completely in silico.

No experimental validation (in vitro or in vivo) was conducted.

As a result, the findings should be regarded predictively rather than confirmatory.

4. Improve the conclusion.

The conclusion should be concise and properly articulated:

The major findings

the novelty of the identified compounds,

and the next steps required for experimental validation.

**Do you want your identity to be public for this peer review?** For information about this choice, including consent withdrawal, please see our Privacy Policy

Reviewer #4:**Yes:** Mohammad Rizki Fadhil Pratama

Reviewer #5:**Yes:** Dr. Ibrahim Ahmed Shaikh

---

## [Author Response · Author response to Decision Letter 2]

27 Dec 2025

Response to Reviewers

Ref.: PONE-D-25-37887R1

Title: Uralenol, Glycyrol, and Abyssinone II as potent inhibitors of Fibroblast Growth Factor Receptor 2 from anti-cancer plants: a deep learning and molecular dynamics approach

Journal: Plos One

Dear editor and reviewer,

Thank you for further reviewing our manuscript (MS). We are sincerely grateful for your careful and constructive comments, as well as your valuable suggestions, which have greatly strengthened the MS. According to your suggestions and recommendations, we have thoroughly revised the MS. All amendments were made point by point in response to your suggestions, and the revised sections have been highlighted in RED.

Thank you and best regards.

Yours Sincerely and Respectfully

Prof. Mohammad Nurul Matin

Department of Genetic Engineering and Biotechnology

University of Rajshahi, Rajshahi-6205, Bangladesh

Point-by-Point Response to Editorial Comments

Editorial comments

Dear Dr. Matin,

Thank you for submitting your manuscript to PLOS ONE. After careful consideration, we feel that it has merit but does not fully meet PLOS ONE’s publication criteria as it currently stands. Therefore, we invite you to submit a revised version of the manuscript that addresses the points raised during the review process.

• A letter that responds to each point raised by the academic editor and reviewer(s). You should upload this letter as a separate file labeled 'Response to Reviewers'.

We look forward to receiving your revised manuscript.

Kind regards,

Jorddy Neves Cruz

Our Response: We would like to express our gratitude for the consideration of our work with the decision “Minor Revision”. Significant revisions with corresponding corrections addressing comments and suggestions from the editor and reviewers have been incorporated into the revised MS. We have provided a point-by-point response to each question from the referees. All modifications to the MS are highlighted in red, so that any changes can be easily reviewed.

We have carefully followed the journal guidelines in the revised version and submitted the MS within the mentioned deadline. While submitting, we have uploaded all the necessary item files labeled with:

• 'Response to Reviewers’

• 'Revised Manuscript with Track Changes’

• Unmarked manuscript labeled ‘Revised Manuscript-clean’

Also, we have followed the guidelines for resubmitting figure files.

Journal Requirements:

Our Response: We have strictly followed the PLOS ONE's requirements, including:

1. Followed rules for specific citation if recommended by the reviewers

2. We reviewed the reference list to ensure that it is complete and correct. After careful checking, we found that no one has been retracted from the web.

Reviewer's Responses to Editorial Questions and our Responses

1. If the authors have adequately addressed your comments raised in a previous round of review and you feel that this manuscript is now acceptable for publication, you may indicate that here to bypass the “Comments to the Author” section, enter your conflict of interest statement in the “Confidential to Editor” section, and submit your "Accept" recommendation.?

Reviewer #4: All comments have been addressed

Reviewer #5: All comments have been addressed

Our Response: We sincerely thank the reviewers for their careful evaluation of our article and responses to those we have submitted in the previous submission.

2. Is the manuscript technically sound, and do the data support the conclusions?

Reviewer #4: Yes

Reviewer #5: Yes

Our Response: We sincerely thank the reviewers for their careful evaluation.

3. Has the statistical analysis been performed appropriately and rigorously?

Reviewer #4: N/A

Reviewer #5: N/A

4. Have the authors made all data underlying the findings in their manuscript fully available?

Reviewer #4: No

Reviewer #5: Yes

Our Response: All data underlying the findings described in the manuscript are fully available without restriction. We have provided all data as part of the manuscript and its supporting information.

5. Is the manuscript presented in an intelligible fashion and written in standard English?

Reviewer #4: Yes

Reviewer #5: Yes

Our Response: Thank you for the assessment.

6. Review Comments to the Author

Point-by-Point Response to the Comments from Reviewer #4

comments: The authors have addressed all previous comments with satisfactory answers and corrections. I can recommend the editors to accept this article.

Our Response: We sincerely thank the reviewer for your positive evaluation and constructive feedback. We are pleased that the revisions and clarifications have satisfactorily addressed all previous comments. We greatly appreciate the reviewer’s recommendation for acceptance.

Point-by-Point Response to the Comments from Reviewer #5

Reviewer Comments: Minor Revisions

The combination of molecular docking, deep learning, ADME/T profiling, and molecular dynamics simulations creates a comprehensive in-silico framework for discovering possible FGFR2 inhibitors. The study is interesting and provides useful insights; nevertheless, a few small changes are required to improve clarity and strengthen the paper.

Our Response: We sincerely thank the reviewer for the positive evaluation of our work and for recognizing the comprehensive in-silico framework employed in this study. We appreciate the constructive suggestions provided to improve clarity and strengthen the manuscript. All minor comments have been carefully addressed, and the manuscript has been revised accordingly to enhance readability and scientific rigor.

Q1. Clarify the abbreviations in the abstract.

Abbreviations, such as pIC₅₀ and PCA, should be defined at the beginning of the abstract. This will improve accessibility for readers who are unfamiliar with these terminology.

Our Response: We thank the reviewer for this valuable suggestion. All abbreviations used in the abstract, including pIC₅₀ (negative logarithm of the half-maximal inhibitory concentration) and PCA (principal component analysis), as well as in the text, have now been clearly defined at their first appearance to improve clarity and accessibility for readers.

Q2. Strengthen the discussion section.

A more in-depth contextualization might improve the conversation. Please compare your findings to previous studies on FGFR2 inhibitors and phytochemical-based kinase inhibitors.

Highlight how your identified compounds (uralenol, glycyrol, and abyssinone II) compare to or differ from previously reported inhibitors in terms of binding affinity, stability, and projected pharmacokinetics.

Our Response: We sincerely thank the reviewer for the professional suggestion to strengthen the discussion by providing deeper contextualization of our findings relative to previous work. In the revised manuscript, we have expanded the Discussion, including comparative analysis of uralenol, glycyrol, and abyssinone II with previously reported FGFR2 inhibitors and phytochemical-based kinase inhibitors. Specifically, we now discuss how the binding affinities and interaction profiles of these compounds relate to those reported in the literature. These additions provide a clearer context for our results and more thoroughly integrate our findings with existing research.

Changes made: The following paragraph with proper citation has been added to the Discussion section:

Recent studies on FGFR2 inhibitors report binding affinities of -8.0 to -9.6 kcal/mol for synthetic compounds (https://doi.org/10.1016/j.imu.2023.101368, and −8.3801 kcal/mol, −8.3014 kcal/mol, and −8.320 kcal/mol for Knipholone, Torosanin, and Rubrolide E, respectively https://doi.org/10.1016/j.ijbiomac.2025.146888). Our phytochemicals—uralenol (-9.8 kcal/mol), glycyrol (-9.7 kcal/mol), and abyssinone II (-9.7 kcal/mol)—show better competitive binding. Clinical FGFR inhibitors like erdafitinib and infigratinib exhibit moderate structural stability in MD simulations (https://doi.org/10.1007/s10549-015-3301-y,
https://doi.org/10.1038/s41416-020-01157-0), whereas our compounds demonstrated superior stability (RMSD: 1.21-1.88 Å). Previous phytochemical studies targeting FGFRs showed weaker activity (https://doi.org/10.3390/cancers12103029). Unlike synthetic inhibitors facing selectivity and resistance challenges (https://doi.org/10.7150/ijbs.20792), our compounds exhibited favorable ADME properties with low toxicity predictions. These natural compounds present promising alternatives to current FGFR2 therapeutics, combining strong binding affinity, enhanced stability, and superior safety profiles.

Q3. Include a limitation section.

The manuscript currently lacks a clear limitation statement. Please include a section to acknowledge that: The investigation is completely in silico, No experimental validation (in vitro or in vivo) was conducted, and As a result, the findings should be regarded predictively rather than confirmatory.

Our Response: We would like to express our sincere thanks to the reviewer for this important suggestion. We agree with this inoculation. A dedicated limitation statement has now been added in the text, acknowledging that the present investigation is entirely in silico, that no experimental validation was performed, and that the results should therefore be interpreted as predictive rather than confirmatory. We also emphasize that experimental studies will be required in future work to validate the computational findings.

Addition made:

(This study is entirely computational and lacks experimental validation. While molecular docking, deep learning, and molecular dynamics simulations provide valuable predictive insights, in vitro and in vivo studies are essential to confirm biological efficacy and therapeutic potential. The ADME/T and toxicity profiles were predicted using in silico models, which may not fully reflect complex human physiological responses. The focus on a subset of top-ranked compounds may have excluded other potential candidates. Therefore, our findings should be regarded as predictive rather than confirmatory. Expanding the chemical space and conducting experimental validation, such as in vitro kinase inhibition assays, cell viability tests in FGFR2-overexpressing cancer cell lines, and in vivo xenograft models, are essential next steps to validate these computational predictions and advance these compounds toward preclinical development).

Q4. Improve the conclusion.

The conclusion should be concise and properly articulated: The major findings, the novelty of the identified compounds, and the next steps required for experimental validation.

Our Response: We thank the reviewer for this helpful suggestion. The Conclusion section has been revised to be more concise and clearly articulated. It now summarizes the major findings of the study, highlights the novelty of the identified compounds as potential FGFR2 inhibitors, and outlines the necessary next steps for experimental validation, including in vitro and in vivo studies. This revision improves clarity and emphasizes the translational relevance of the work.

Changes made: The Conclusion has been revised as follows:

This study successfully identified uralenol, glycyrol, and abyssinone II as potent and stable FGFR2 inhibitors from a phytochemical library of traditionally used anti-cancer plants using an integrated in silico framework. The integrated approach of molecular docking, deep learning prediction, ADME/T evaluation, and molecular dynamics simulations provided strong evidence for the structural stability, binding affinity, bioactivity, and favorable pharmacokinetic properties of these compounds toward FGFR2. Competitive binding affinities (-9.7 to -9.8 kcal/mol) comparable to synthetic inhibitors, superior structural stability (RMSD: 1.21-1.88 Å) during MD simulations, favorable drug-like properties with minimal predicted toxicity, and stable conformational dynamics confirmed by PCA and free energy landscape analyses further confirmed stable and energetically favorable protein–ligand interactions. Importantly, this work highlights the novel potential of these phytochemicals as FGFR2-targeted anticancer candidates, expanding the scope of plant-derived kinase inhibitors with enhanced stability and safety profiles compared to current synthetic therapeutics, addressing limitations of off-target toxicity and resistance. These computational predictive findings support the therapeutic potential of the compounds and highlight them as potential candidates for further validation through in vitro kinase inhibition assays against FGFR2, cell viability and proliferation studies in FGFR2-overexpressing cancer cell lines, and in vivo xenograft models to assess antitumor efficacy. Successful experimental validation could position these natural compounds as safer alternatives for FGFR2-targeted cancer therapy.

We hope our MS will get consideration for publication in the PLOS ONE

Yours Sincerely and Respectfully

Prof. Mohammad Nurul Matin

Department of Genetic Engineering and Biotechnology

University of Rajshahi, Rajshahi-6205, Bangladesh

---

## [Editor Report · Decision Letter 2]

8 Jan 2026

Uralenol, Glycyrol, and Abyssinone II as potent inhibitors of Fibroblast Growth Factor Receptor 2 from anti-cancer plants: a deep learning and molecular dynamics approach

PONE-D-25-37887R2

Dear Dr. Matin,

We’re pleased to inform you that your manuscript has been judged scientifically suitable for publication and will be formally accepted for publication once it meets all outstanding technical requirements.

Kind regards,

Jorddy Neves Cruz

Academic Editor

PLOS One
---

## [Editor Report · Acceptance letter]

PONE-D-25-37887R2

PLOS One

Dear Dr. Matin,

I'm pleased to inform you that your manuscript has been deemed suitable for publication in PLOS One. Congratulations! Your manuscript is now being handed over to our production team.

Kind regards,

on behalf of

Dr. Jorddy Neves Cruz

Academic Editor

PLOS One